# Draft Genomes of Two *Artocarpus* Plants, Jackfruit (*A. heterophyllus*) and Breadfruit (*A. altilis*)

**DOI:** 10.3390/genes11010027

**Published:** 2019-12-24

**Authors:** Sunil Kumar Sahu, Min Liu, Anna Yssel, Robert Kariba, Samuel Muthemba, Sanjie Jiang, Bo Song, Prasad S. Hendre, Alice Muchugi, Ramni Jamnadass, Shu-Min Kao, Jonathan Featherston, Nyree J. C. Zerega, Xun Xu, Huanming Yang, Allen Van Deynze, Yves Van de Peer, Xin Liu, Huan Liu

**Affiliations:** 1BGI-Shenzhen, Shenzhen 518083, China; sunilkumarsahu@genomics.cn (S.K.S.); liumin4@genomics.cn (M.L.); jiangsanjie@bgi.com (S.J.); songbo@genomics.cn (B.S.); xuxun@genomics.cn (X.X.); yanghm@genomics.cn (H.Y.); 2State Key Laboratory of Agricultural Genomics, BGI-Shenzhen, Shenzhen 518083, China; 3Center for Microbial Ecology and Genomics (CMEG), Department of Biochemistry, Genetics and Microbiology, University of Pretoria, Pretoria, Hatfield 0028, South Africa; aejysselansie@gmail.com (A.Y.); yvpee@psb.vib-ugent.be (Y.V.d.P.); 4African Orphan Crops Consortium, World Agroforestry Centre (ICRAF), Nairobi 00100, Kenya; R.Kariba@cgiar.org (R.K.); S.MUTHEMBA@cgiar.org (S.M.); P.Hendre@cgiar.org (P.S.H.); a.muchugi@cgiar.org (A.M.); R.JAMNADASS@CGIAR.ORG (R.J.); avandeynze@ucdavis.edu (A.V.D.); 5Department of Plant Biotechnology and Bioinformatics, Ghent University, Ghent, Zwijnaarde 9052, Belgium; shkao@psb.vib-ugent.be; 6Center for Plant Systems Biology, VIB, Ghent, Zwijnaarde 9052, Belgium; 7Biotechnology Platform, Agricultural Research Council, Pretoria 0110, South Africa; featherstonj@arc.agric.za; 8Chicago Botanic Garden, Negaunee Institute for Plant Conservation Science and Action, Glencoe, IL 60022, USA; n-zerega@northwestern.edu; 9Plant Biology and Conservation, Northwestern University, Evanston, IL 60208, USA; 10Seed Biotechnology Center, University of California, 1 Shields Ave, Davis, CA 95616, USA; 11Department of Biology, University of Copenhagen, DK-1165 Copenhagen, Denmark

**Keywords:** jackfruit, breadfruit, *A. heterophyllus*, *A. altilis*, starch synthesis

## Abstract

Two of the most economically important plants in the *Artocarpus* genus are jackfruit (*A. heterophyllus* Lam.) and breadfruit (*A. altilis* (Parkinson) Fosberg). Both species are long-lived trees that have been cultivated for thousands of years in their native regions. Today they are grown throughout tropical to subtropical areas as an important source of starch and other valuable nutrients. There are hundreds of breadfruit varieties that are native to Oceania, of which the most commonly distributed types are seedless triploids. Jackfruit is likely native to the Western Ghats of India and produces one of the largest tree-borne fruit structures (reaching up to 45 kg). To-date, there is limited genomic information for these two economically important species. Here, we generated 273 Gb and 227 Gb of raw data from jackfruit and breadfruit, respectively. The high-quality reads from jackfruit were assembled into 162,440 scaffolds totaling 982 Mb with 35,858 genes. Similarly, the breadfruit reads were assembled into 180,971 scaffolds totaling 833 Mb with 34,010 genes. A total of 2822 and 2034 expanded gene families were found in jackfruit and breadfruit, respectively, enriched in pathways including starch and sucrose metabolism, photosynthesis, and others. The copy number of several starch synthesis-related genes were found to be increased in jackfruit and breadfruit compared to closely-related species, and the tissue-specific expression might imply their sugar-rich and starch-rich characteristics. Overall, the publication of high-quality genomes for jackfruit and breadfruit provides information about their specific composition and the underlying genes involved in sugar and starch metabolism.

## 1. Introduction

The family Moraceae contains at least 39 genera and approximately 1100 species [1,2,3]. Species diversity of the family is primarily centered in the tropics with variation in inflorescence structures, pollination forms, breeding systems, and growth forms [2]. Within the Moraceae family, the genus *Artocarpus* is comprised of approximately 70 species [2,4]. The most recent evidence indicates that Borneo was the center of diversification of the *Artocarpus* genus and that species diversified throughout South and Southeast Asia [2]. All members of the genus have unisexual flowers and produce exudate from laticifers. Inflorescences consist of up to thousands of tiny flowers, tightly packed and condensed on a receptacle [2]. In most species, the perianths of adjacent female flowers are partially to completely fused together and develop into a highly specialized multiple fruit called a syncarp, which is formed by the enlargement of the entire female head. Syncarps of different species range in size from a few centimeters in diameter to over half a meter long in the case of jackfruit [2,5]. Many *Artocarpus* species are important food sources for forest fauna, and about a dozen species are important crops in the regions where they are from [2,6].

Jackfruit (*A. heterophyllus* Lam.) is thought to have originated in in the Western Ghats of India and is cultivated as an important food source across the tropics. It is monecious, and thought to be pollinated by gall midges [7]. In some areas it is propagated mainly by seeds [8], however, clonal propagation via grafting is increasing in areas where it is grown for commercial use [9]. On average it contains more than 100 seeds per fruit with viability of less than a month [10,11]. The male flowers are tiny and clustered on an oblong receptacle, typically 5–10 cm in length. Limited studies exist on the range of cultivated varieties of jackfruit, but they are often grouped into two main types, varieties with edible fleshy perianth tissue (often referred to as “flakes”) that are either (a) small, fibrous, soft, and spongy, or (b) larger, less sweet crisp fruit [10,12]. The latter type is often more commercially important.

Breadfruit (*A. altilis* [Parkinson] Fosberg) is most likely derived from the progenitor species *A. camansi* Blanco, which is native to New Guinea [10,13]. As humans migrated and colonized the islands of Remote Oceania, indigenous people selected and cultivated varieties from the wild ancestor over thousands of years [13], giving rise to hundreds of cultivated varieties [10,13,14,15]. Cultivated varieties were traditionally propagated clonally by root cuttings but can now be commercially propagated by tissue culture [16,17]. Among the hundreds of varieties, some are diploid (2n = 2x = ~56) and may produce seeds, while other varieties are seedless triploids (3n = 2x = ~84), and still others are of hybrid origin with another species, *A. mariannensis* Trécul [13,18,19,20]. A small subset of the triploid diversity is what has been introduced outside of Oceania [19,21].

To diversify the global food supply, enhance agricultural productivity, and eradicate malnutrition, it is necessary to focus on crop improvement of plants that are utilized in rural societies as a local source of nutrition and sustenance. This study is part of the African Orphan Crops Consortium (AOCC), an international public-private partnership. A goal of this global initiative is to sequence, assemble, and annotate the genomes of 101 traditional African food crops [22,23]. Both breadfruit and jackfruit are nutritious [24,25,26,27] and have the potential to increase food security, especially in tropical areas. Until now limited genomic information has been available for the *Artocarpus* genus as a whole. Microsatellite markers have been used to characterize cultivars and wild relatives of breadfruit [8,19,21,28], jackfruit [29], and other *Artocarpus* crop species [6,30,31]. Additionally, an assembled and annotated reference transcriptome of *A. altilis* has been generated [20]. Twenty-four transcriptomes of breadfruit and its wild relatives revealed signals of positive selection that may have resulted from local adaptation or natural selection [20]. Finally, a low coverage whole genome sequence has been published for *A. camansi* [32], but full genome sequences for jackfruit and breadfruit are still not available. Here, we report high-quality annotated draft genome sequences for both jackfruit and breadfruit. The results help explain their energy-dense fruit composition and the underlying genes involved in sugar and starch metabolism.

## 2. Materials and Methods

### 2.1. Sample Collection, NGS Library Construction, and Sequencing

Genomic DNA was extracted from fresh leaves of *A. heterophyllus* (ICRAFF 11314) and *A. altilis* (ICRAFF 11315), grown at the World AgroForestry (ICRAF) campus in Kenya, using a modified CTAB method [33]. Extracted DNA was used to construct four paired-end libraries (170, 350, 500, and 800 bp) and four mate-pair libraries (2, 6, 10, and 20 Kb) following the standard protocols provided by Illumina (San Diego, CA, USA). Subsequently, the sequencing was performed on a HiSeq 2000 platform (Illumina, San Diego, CA, USA) using a whole genome shotgun sequencing strategy. To improve the data quality, the poor quality reads were filtered using SOAPfilter (v2.2) [34] with the following parameters: (1) low-quality bases with Q = below 7 and 15 for *A. altilis* and *A. heterophyllus,* respectively were trimmed from forward and reverse reads; (2) reads with ≥30% low quality bases (quality score ≤ 15); (3) reads with ≥10% uncalled (“N”) bases; (4) reads with adapter contamination or PCR duplicates; (5) reads with undersized insert sizes were discarded. Finally, more than 100 high-quality reads were obtained for each species (see Appendix A).

For transcriptome sequencing, the RNA was extracted from different tissues of *A. altilis* (various leaf stages, leaf buds, and roots) and *A. heterophyllus* (various leaf stages, leaf bud, stem, bark, roots, germinated seed, and seedling). The RNA was extracted using the PureLink RNA Mini Kit (Thermo Fisher Scientific, Carlsbad, CA, USA) according to the manufacturer’s instructions. For each sample, RNA libraries were constructed by following the TruSeq RNA Sample Preparation Kit (Illumina, San Diego, CA, USA) manual, and were then sequenced on the Illumina HiSeq 2500 platform (paired-end, 100-bp reads), generating more than 47 Gb of sequence data for each species. Data were then filtered using a similar criterion as used to filter DNA NGS data, with a slight modification: (1) reads with ≥10% low-quality bases (quality score ≤ 15) were removed; and (2) reads with ≥5% uncalled (“N”) bases were removed (see Appendix A). All of the transcriptome data were compiled, and the combined version was used to check the completeness of the whole genome sequence assembly.

### 2.2. Evaluation of Genome Size

Clean reads of the paired-end libraries (170, 250, and 500 bp) were used to estimate the genome size by k-mer frequency distribution and heterozygosity analysis. The genome size was estimated based on the following formula: G = N × (L − 17 + 1)/K_depth(1)
where N represents the number of used reads, L represents the read length, K represents the k-mer value in the analysis, and K_depth refers to the location of the main peak in the distribution curve [35]. The heterozygosity was evaluated by the GCE software [36].

### 2.3. De Novo Genome Assembly

The de novo genome assembly tool, Platanus (Platanus, RRID: SCR_015531) [37], was used to construct the contigs and scaffolds in three steps: contig assembling, scaffolding and gap closing. In contig assembling, paired-end libraries ranging from 170 to 800 bp were used with the parameters “-d 0.5 -K 39 -u 0.1 -m 300”. In the scaffolding step, paired-end and mate-pair information were used to with parameters “-u 0.1”. Lastly, the paired-end reads were used for gap closing using GapCloser version 1.12 (GapCloser, RRID: SCR_015026) [34], with the parameters “-l 150 -t 32 -p 31”.

### 2.4. Genome Assembly Evaluation

The genome assembly completeness was assessed using BUSCO (Benchmarking Universal Single-Copy Orthologues), version 3.0.1 (BUSCO, RRID: SCR_015008) [38]. Next, the unigenes generated by Bridger software [39] from the transcriptome data of each species were aligned to the assembled genomes using BLAT (BLAT, RRID: SCR_011919) [40] with default parameters. In order to confirm the accuracy of the assembly, some of the paired-end libraries (170, 250, and 350 bp) were aligned to the assembled genomes, and the sequencing coverage was calculated using SOAPaligner, version 2.21 (SOAPaligner/soap2, RRID: SCR_005503) [41].

We calculated the GC content and average depth with 10 kb non-overlapping windows. The distribution of GC content indicated a relative pure single genome without contamination or GC bias (Appendix A). Moreover, the GC content of *A. altilis* and *A. heterophyllus* genomes were also compared with three rosids species (*Fragaria vesca*, *Malus domestica*, and *Morus notabilis*).

### 2.5. Repeat Annotation

Repetitive sequences were identified by using RepeatMasker (version 4-0-5) [42], with a combined library consisting of the Repbase library and a custom library obtained through careful self-training. The custom library was composed of three parts: the MITE, LTR, and an extensive library, which were constructed as described below. First of all, the library of miniature inverted-repeat transposable elements (MITEs) was created by annotation using MITE-hunter [43] with default parameters. Secondly, the library of long terminal repeats (LTR) was constructed using LTRharvest [44] integrated in Genometools (version 1.5.8) [45] with parameters “-minlenltr 100 -maxlenltr 6000 -mindistltr 1500 -maxdistltr 25,000 -mintsd 5 -maxtsd 5 -similar 90 -vic 10” to detect LTR candidates with lengths of 1.5 kb to 25 kb, including two terminal repeats ranging from 100 bp to 6000 bp with ≥ 85% similarity. In order to improve the quality of the LTR library, we used several strategies to filter the candidates. As intact PPT (poly purine tract) or PBS (primer binding site) was necessary to define LTR, we subsequently used LTRdigest [46] with a eukaryotic tRNA library [47] to identify these features, and then removed the elements without appropriate PPT or PBS locations. Subsequently, to remove contamination, like local gene clusters and tandem local repeats, 50 bp of flanking sequences on both sides of each LTR candidate were aligned using MUSCLE (MUSCLE, RRID:SCR_011812) [48] with default parameters: if the identity ≥ 60%, the candidate was taken as a false positive and removed. LTR candidates which nested with other types of elements were also removed. Exemplars for the LTR library were extracted from the filtered candidates using a cutoff of 80% identity in 90% of sequence length. Furthermore, the regions annotated as LTRs and MITEs in the genome were masked, and then put into RepeatModeler version 1-0-8 RepeatModeler, RRID: SCR_015027) to predict other repetitive sequences for the extensive library.

Finally, the MITE, LTR and extensive libraries were integrated into the custom library, which was combined with the Repbase library and then taken as the input for RepeatMasker to identify and classify repetitive elements genome wide.

### 2.6. Gene Prediction

Repetitive regions of the genome were masked before gene prediction. Based on the RNA, homologous and de novo prediction evidence, the protein-coding genes were identified using the MAKER-P pipeline (version 2.31) [49]. For RNA evidence, the clean transcriptome reads were assembled into inchworms using Trinity version 2.0.6 [50], and then fed to MAKER-P as EST evidence. For homologous evidence, the protein sequences from four relative species in the rosids (*F. vesca*, *M. domestica*, *M. notabilis*, *Prunus persica, Ziziphus jujuba*) were downloaded and provided as protein evidence.

For de novo prediction evidence, a series of training attempts were made to optimize different ab initio gene predictors. At first, a set of transcripts were generated by a genome-guided approach using Trinity with parameters “--full_cleanup --jaccard_clip --genome_guided_max_intron 10,000 --min_contig_length 200”. The transcripts were then mapped back to the genome using PASA (version 2.0.2) [51] and a set of gene models with real gene characteristics (e.g., size and number of exons/introns per gene, features of spicing sites) was generated. The complete gene models were picked for training Augustus [52]. Genemark-ES (version 4.21) [53] was self-trained with default parameters. The first round of MAKER-P was run based on the evidence above with default parameters except “est2genome” and “protein2genome” set to “1”, yielding only RNA- and protein-supported gene models. SNAP [54] was then trained with these gene models. Default parameters were used to run the second and final round of MAKER-P, producing final gene models. 

Furthermore, non-coding RNA genes in the *A. altilis* and *A. heterophyllus* genomes were also annotated. The BLAST tool was employed to search ribosomal RNA (rRNA) against the *A. thaliana* rRNA database, and to search microRNAs (miRNA) and small nuclear RNA (snRNA) against ther Rfam database (Rfam, RRID: SCR_004276) (release 12.0) [55]. tRNAscan-SE (tRNAscan-SE, RRID: SCR_010835) [56] was used to scan transfer RNA (tRNA) in the genome sequences. 

### 2.7. Functional Annotation of Protein-Coding Genes

Functional annotation of protein-coding genes was based on sequence similarity and domain conservation by aligning translated coding sequences to public databases. The protein-coding genes were first queried against protein sequence databases, such as KEGG (KEGG, RRID: SCR_012773) [57], NR database (NCBI), COG [58], SwissProt, and TrEMBL [59] for best-matches using BLASTP with an E-value cut-off of 1 × 10^−5^. Secondly, InterProScan 55.0 (InterProScan, RRID: SCR_005829) [60] was used as an engine to identify the motif and domain-based on Pfam (Pfam, RRID: SCR_004726) [61], SMART (SMART, RRID: SCR_005026) [62], PANTHER (PANTHER, RRID: SCR_004869) [63], PRINTS (PRINTS, RRID: SCR_003412) [64] and ProDom (ProDom, RRID: SCR_006969) [65,66].

### 2.8. Ks-Distribution Analysis

The coding sequences and annotations for *Morus notabilis* were downloaded from the NCBI, reference RefSeq assembly accession GCF_000414095.1 [66]. The coding sequences and annotations for *Ziziphus jujube* [67] were downloaded from the Plaza4 database [68]. The headers of the fasta files, as well as the ninth columns of the gff3 files were edited to make the datasets compatible with the software packages used for downstream analysis.

Ks-distribution analyses were performed, using the wgd-package [69]. For each species, the paranome was obtained by performing an all-against-all BlastP [70], with MCL clustering [71]. Codon multiple sequence alignment was done using MUSCLE [48]. Ks-distributions were constructed using codeml from the PAML4 package [72] and Fast-Tree [73] for inferring phylogenetic trees used in the node weighting procedure, other software used by the wgd. Thereafter, i-ADHoRe [74] was used to get anchor-point distributions and produce dot-plots. Lastly, Gausian mixture modes were fitted using 1–5 components.

### 2.9. One vs. One Synteny

One-vs.-one synteny analysis was performed for pairs of the above-mentioned species, using the “work-flow 2” script that is part of the wgd-package [69]. In order to compare *A. altilis*, *A. heterophyllus*, we use MCScanX to detect the synteny and collinearity, and MCscan (Python version) for the visualization.

### 2.10. Gene Family Construction

Protein and nucleotide sequences from *A. altilis*, *A. heterophyllus*, and seven species (*A. thaliana*, *F. vesca*, *M. domestica, M. notabilis, P. mume, P. persica, Z. jujuba*) were retrieved to construct gene families using OrthoMCL software [75] based on an all-versus-all BLASTP alignment with an E-value cutoff of 1 × 10^−5^.

### 2.11. Collinearity Analysis

Collinearity of the largest orthologous scaffolds were determined and plotted using MCscan and the JCVI utility libraries v0.9.14 [76,77].

### 2.12. Phylogenetic Analysis and Divergence Time Estimation

We identified 486 single-copy genes in the nine species, and subsequently used them to build the phylogenetic tree. Coding DNA sequence (CDS) alignments of each single-copy family were constructed following the protein sequence alignment with MUSCLE (MUSCLE, RRID: SCR_011812) [48]. The aligned CDS sequences of each species were then concatenated to a super gene sequence. The phylogenetic tree was constructed with PhyML-3.0 (PhyML, RRID: SCR_014629) [78] with the HKY85+ Gamma substitution model on extracted four-fold degenerate sites. Divergence time was calculated using the Bayesian relaxed molecular clock approach using MCMCTREE in PAML (PAML, RRID: SCR_014932) [72], based on the published calibration times (divergence between *Arabidopsis thaliana* and Rosales was 108-109 Mya, divergence between *P. mume* and *P. persica* was 24–72 Mya) [66]. The divergence time between *M. notabilis* and *Artocarpus* was predicted to be 61.8 (54.1–76.0) Mya (Figure 1A). Subsequently, to study gene gain and loss, CAFE (CAFE, RRID:SCR_005983) [79] was employed to estimate the universal gene birth and death rate λ (lambda) under a random birth and death model with the maximum likelihood method. The results for each branch of the phylogenetic tree were estimated (Figure 1A). Enrichment analysis on GO and the pathway of genes in expanded families in the *Artocarpus* lineage were also calculated.

### 2.13. Identification of Starch Biosynthesis-Related Genes

Using the amino acid, starch biosynthesis-related genes in soybean as bait, we performed an ortholog search in *A. altilis*, *A. heterophyllus, M. notabilis, Z. jujuba, P. mume, P. persica*, *F. vesca, M. domestica*, and *A. thaliana*. 

## 3. Results and Discussion

### 3.1. Genome Sequencing and Assembly

A total of eight libraries were constructed including four short-insert libraries (170 bp, 350 bp, 500 bp, and 800 bp) and four mate-pair libraries (2 kb, 5 kb, 10 kb, and 20 kb) for Illumina Hiseq2000 sequencing. In total, 273 Gb and 227 Gb of raw data was generated from *A. heterophyllus* and *A. altilis*, respectively (Appendix A). We used the GCE software to evaluate the heterozygosity, and the results showed that the heterozygous ratio is 1.13% and 0.911% for *A. altilis* and *A. heterophyllus*, respectively. The K-mer distributions of *A. altilis* and *A. heterophyllus* showed two distinct peaks (Appendix A), the first peak was the heterozygous peak, the second peak was the homozygous peak, where the second peak was confirmed as the main one for each of the species. Based on K-mer frequency methods [36], the *A. heterophyllus* and *A. altilis* genomes were estimated to be 1005 MB and 812 Mb, respectively (Appendix A). The genome sizes of *A. altilis* and *A. heterophyllus* were relatively close to the genome size of species in the genus *Artocarpus* based on existing data in the 1C-values database of 1.2 pg.

Using the SOAPdenovo2 program [41], all of the *A. heterophyllus* high-quality reads were assembled into 108,267 scaffolds, totaling 982 Mb (Table 1). The N50s of contigs and scaffolds were 27 kb and 548 kb, with the longest being 255 kb and 3.1 Mb, respectively (Table 1, Appendix A). Similarly, for *A. altilis*, the N50s of contigs and scaffolds were 17 kb and 1.5 Mb with the longest being 174 kb and 7.4 Mb respectively (Table 1, Appendix A). These results indicate the high quality of the assemblies for both species. The GC content of the *A. heterophyllus* and *A. altilis* genomes were 32.9% and 32.3%, respectively. The GC depth graphs and distributions indicated there was no contamination in the genome assemblies (Appendix A). 

Evaluation of the quality and completeness of the draft genome assembly was done with the Benchmarking Universal Single-Copy Orthologs (BUSCO) datasets [38]. Of the total of 1440 BUSCO ortholog groups searched in the *A. heterophyllus* assembly, 932 (64.7%) BUSCO genes were “complete single-copy”, 437 (30.3%) were “complete duplicated”, 15 (1%) were “fragmented”, and 56 (4%) were “missing” (Table 2). Similarly, in *A. altilis*, 988 (68.6%) BUSCO genes were “complete single-copy”, 383 (26.6%) were “complete duplicated”, 14 (~1%) were “fragmented”, and 55 (3.8%) were “missing” (Table 2), suggesting that the quality of the genome assembly is high. From the 1440 core Embryophyta genes, 1371 (95.2%) and 1369 (95.1%) were identified in the *A. altilis* and *A. heterophyllus* assemblies, respectively (Table 2). We observed a significant difference in the number of duplicated core genes in *A. altilis* and *A. heterophyllus* (Table 2), which might be ascribed to the genome duplication in these species. The results also indicated that the assembly covered more than 90% of the expressed unigenes, (Table 3). As expected, after the comparative GC content analysis the close peak positions showed *A. altilis*, *A. heterophyllus*, and *M. notabilis* are closer than other species in GC content (Appendix A).

### 3.2. Gene Annotation

A combination of de novo and homology-based methods (using transcript data as evidence) were used to identify repeat sequences. We found that up to 51.01% of the *A. heterophyllus* and 52.04% of the *A. altilis* assembled sequences were repeat sequences, comprised mostly of transposable elements and tandem repeats. Interestingly, the amounts of these elements were higher than what is observed in orange (20%, 367 Mb) [80], peach (29.6%, 265 Mb) [81], pineapple (38.3%, 526 Mb) [82], and others (Table 4). This is consistent with the finding that larger fruit tree genomes often retained higher percentages of repetitive elements compared to the smaller fruit tree genomes [83]. Among the repetitive sequences, 37.0% and 46.0% were of the long terminal repeat (LTR) type, respectively (Table 4), indicating LTRs are the most abundant transposable elements in *A. heterophyllus* and *A. altilis* genomes.

Using a comprehensive annotation strategy, we annotated a total of 35,858 *A. heterophyllus* genes and 34,010 *A. altilis* genes (Table 5). This was close to the number of genes (39,282) predicted in *Dimocarpus longan*, an exotic round to oval Asian fruit [83]. The average *A. heterophyllus* gene length was 3472 bp, the average length of the coding sequence (CDS) was 1241 bp, and the average number of exons per gene was 5.5 (Table 5, Appendix A). We predicted a total of 466 rRNA, 159 miRNA, 1554 snRNA genes and 713 tRNA in *A. altilis*; and a total of 2706 rRNA, 168 miRNA, 1005 snRNA genes and 689 tRNA in *A. heterophyllus* (Table 6). Of 35,858 *A. heterophyllus* protein-coding genes, 35,076 (97.8%) had Nr homologs, 34,968 (97.5%) had TrEMBL homologs, 27,632 (77.1%) had InterPro homologs and 27,741 (77.4%) had SwissProt homologs (Table 7). Similar to *A. heterophyllus*, the average *A. altilis* gene size was 3545 bp, the average length of the CDS was 1253 bp, and the average number of exons per gene was 5.5 (Table 5). Of 34,010 *A. altilis* protein-coding genes, 33,353 (98.1%) had Nr homologs, 33,240 (97.7%) had TrEMBL homologs, 26,422 (77.7%) had InterPro homologs, and 26,689 (78.5%) had SwissProt homologs (Table 7). BUSCO evaluation showed that more than 89% of 1440 core genes were complete, suggesting an acceptable gene annotation for *A. altilis* and *A. heterophyllus* genomes (Appendix A)

### 3.3. Gene Family Evolution and Comparison

Orthologous clustering analysis was conducted with the *A. altilis* and *A. heterophyllus* genomes following comparison with seven other plant genomes: *A. thaliana*, *F. vesca*, *M. domestica, M. notabilis, P. mume, P. persica,* and *Z. jujuba*. A Venn diagram shows that *A. altilis*, *A. heterophyllus*, *A. thaliana*, *M. notabilis*, and *Z. jujuba* contain a core set of 9462 gene families in common; there were 1028 orthologous families shared by three Moraceae species; while 329 gene families containing 515 genes were specific to *A. altilis*; and 420 gene families containing 907 genes were specific to *A. heterophyllus* (Figure 1C).

Of the 35,845 protein-coding genes in the *A. heterophyllus* genome, 28,969 were grouped into 15,768 gene families (of which 242 were *A. heterophyllus*-unique families) (Figure 1B, Appendix A). Of the 33,986 *A. altilis* protein-coding genes, 27,354 were grouped into 15,614 gene families (of which 136 were *A. altilis*-unique families) (Figure 1B, Appendix A).

Phylogenetic analysis showed that *A. heterophyllus* and *A. altilis* were more closely related to mulberry than to Jujube (Figure 1A), further supporting a previous phylogeny of *Artocarpus* [2]. CAFE [79] was used to identify gene families that had potentially undergone expansion or contraction. We found a total of 2822 expanded gene families and 1497 contracted families in *A. heterophyllus*, as well as 2034 expanded and 1800 contracted families in *A. altilis* (Figure 1A). The genes in the expanded and contracted families were assigned to Kyoto Encyclopedia of Genes and Genomes (KEGG) pathways [84]. The *A. heterophyllus*-expanded gene families were remarkably enriched in metabolism related pathways/functions, including starch and sucrose metabolism (ko00500, *p* = 0.003), glycan degradation (ko00511, *p* = 0.007), glycolysis/gluconeogenesis (ko00010, *p* = 0.016), and others (Appendix A). KEGG enrichment analysis of *A. altilis* revealed that pathways associated with photosynthesis, such as carbon fixation in photosynthetic organisms (ko00710, *p* = 0.017), other types of O-glycan biosynthesis (ko00514, *p* = 0.018), and photosynthesis (ko00195, *p* = 0.006) were particularly enriched (Appendix A).

Collinearity between the largest orthologous scaffolds were determined (Appendix A) and the result indicates conserved shared synteny. In order to determine whether there is any evidence for whole genome duplications in *A. heterophyllus* and *A. altilis*, the distance–transversion rates at four-fold degenerate sites (4DTv) was calculated (Figure 1D, Appendix A). Two 4DTv values that peaked at 0.07 and 0.08 for orthologs between *A. heterophyllus*, and between *A. altilis*, respectively, which highlighted the recent whole-genome duplication of these two species. The results of the Ks distributions mostly corroborate the findings of the 4DTv analysis. The results suggest that the whole genome duplication event was shared by *A. altilis* and *A. heterophylus*. Their divergence is recent, as suggested by the overlap of their WGD peaks (Figure 2), meaning that they have equal substitution, duplication, and loss rates. Thus, for further analysis (one-vs.-one synteny with the close relatives *M. notabilis* and *Z. jujube)*, only *A. altilis* was used. These results suggest that the *Artocarpus* genome duplication event occurred after divergence from the common ancestor they share with *M. notabilis* (Figure 3), thus, between 62 and 10 MYA. A total of 33,614 gene pairs were identified in 2694 syntenic blocks.

### 3.4. Gene Family Expansion and Tissue Specific Expression of Starch Synthesis-Related Genes

The copy number of starch synthesis related genes were compared between *A. heterophyllus*, *A. altilis*, closely related species, as well as some other starch-rich plant species (Figure 4). We observed a remarkable copy number expansion of the *UGD1* gene in *A. heterophyllus* compared with the other species. The enzyme encoded by *UGD1*, catalyzes the conversion of Glucose-1-P into UDP-GlcA, thereby stalling the starch synthesis process [85] (Figure 4). Interestingly, the tissue-specific expression pattern of *UGD1* contrasts with other starch synthesis genes in *A. heterophyllus* (Figure 5A). For instance, in *A. heterophyllus* there is a suppression of UDPG transcription in the stem, while the other starch biosynthesis genes are activated. However, differential expression of *UGD1* was not shown in *A. altilis* (Figure 5B). This unusual expression pattern of *UGD1*, as well as the gene copy number expansion, might lead to the failure of starch accumulation in *A. heterophyllus* rather than *A. altilis*. However, this needs to be further validated by real-time qPCR for confirmation of the tissue-specific expression. For the GO enrichment, expansion of gene families were related to small molecule binding or single organism signaling (Appendix A) in *A. altilis*. Moreover, there were some expansion of gene families related to molecule binding, reproductive process, and cellular response to stimulus in *A. heterophyllus*. Gene families belonging to expanded pathways in *A. altilis* were mainly related to plant-pathogen interaction, lysine biosynthesis or photosynthesis. In contrast, the gene families that were expanded in *A. heterophyllus* belonged to pathways involving secondary metabolite biosynthesis, phenylpropanoid biosynthesis, and fatty acid metabolism. In contrast, the biosynthesis of secondary metabolites, phenylpropanoid biosynthesis, and fatty acid metabolism were enriched in the expanded gene families in *A. heterophyllus* (Appendix A).

## 4. Conclusions

Here, we report the characterization of the genomes of jackfruit (*A. heterophyllus*) and breadfruit (*A. altilis*). The publication of these high-quality draft genomes and annotations may provide plant breeders and other researchers with useful information regarding trait biology and their subsequent improvement. In particular, we highlight genes unique to *A. heterophyllus* and *A. altilis* due to their high sugar and starch content, respectively, which are desirable characteristics in these edible plants. The information provided in the draft genome annotations can be used to accelerate genetic improvement of these crops. The availability of these genomes on the AOCC ORCAE platform (https://bioinformatics.psb.ugent.be/orcae/aocc) will enable various stakeholders to access and improve the annotations of these genomes.

## Figures and Tables

**Figure 1 genes-11-00027-f001:**
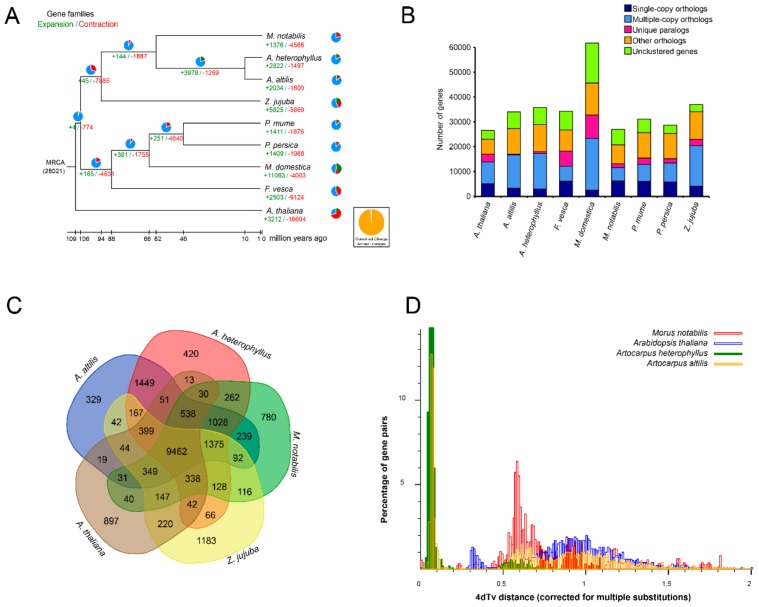
Phylogenetic and evolutionary analysis. (**A**,**B**) Gene conservation and gene family expansion and contraction in *A. heterophyllus* and *A. altilis*. The scale bar indicates 10 million years. The values at the branch points indicate the estimates of divergence time (mya), while the green numbers show the divergence time (million years ago, Mya), and the red nodes indicate the previously published calibration times. (**C**) The distribution of gene families among the model species and *Artocarpus* genus. (**D**) Distribution of 4DTv distance between collinearity gene pairs among *A. heterophyllus*, *A. altilis*, *M. notabilis*, and *Arabidopsis thaliana*.

**Figure 2 genes-11-00027-f002:**
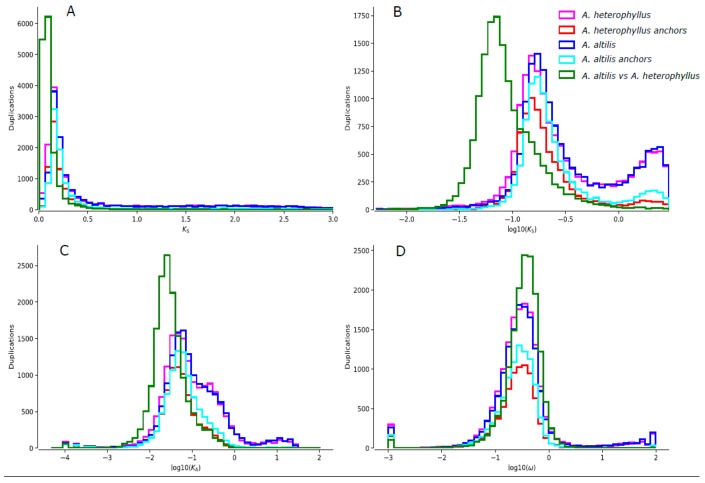
A graph showing the *A. heterophyllus* Ks distributions (pink) and the Ks distributions of its anchor pairs (red), *A altilis* Ks distributions (dark blue) and the distributions of its anchor pairs (light blue) overlaid with the Ks distributions of the one-to-one orthologs of *A. heterophyllus*, and *A. altilis* (green). (**B**) log transformed version of (**A**). (**C**,**D**) are the log transformed Ka and ω distributions, respectively.

**Figure 3 genes-11-00027-f003:**
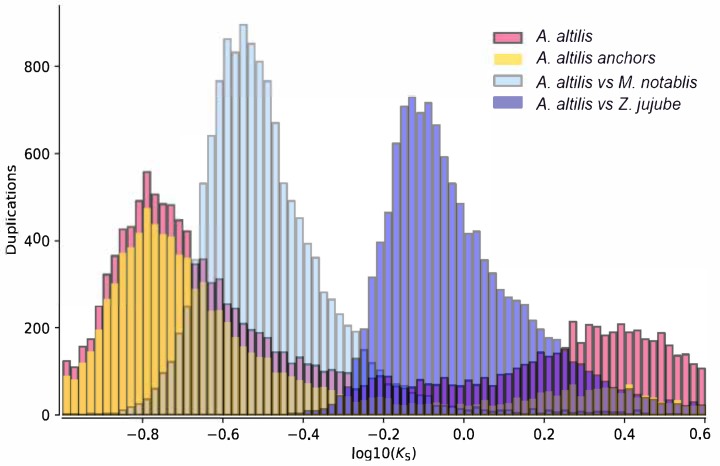
Ks distribution (dark pink) and anchor pair Ks distribution (yellow) of *A. altilis* in overlay with the results of whole paranome distributions between *A. altilis* and *M. notabilis* (light blue) and *A. altilis* and *Z. jujube* (dark blue).

**Figure 4 genes-11-00027-f004:**
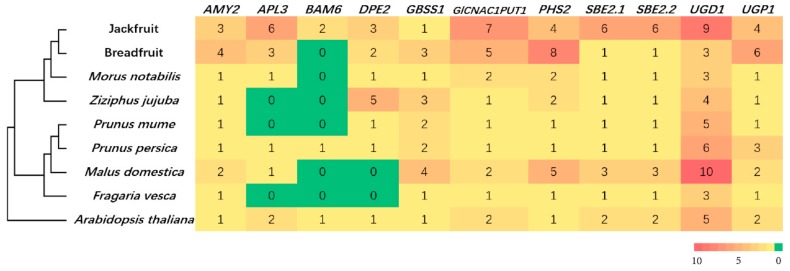
Copy number expansion of starch synthesis-related genes in *A. heterophyllus* and *A. altilis*.

**Figure 5 genes-11-00027-f005:**
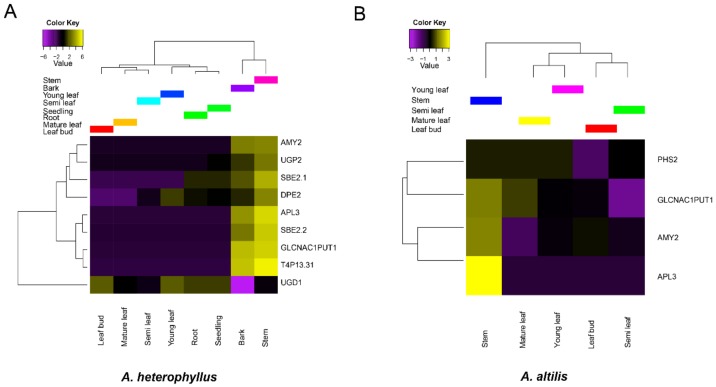
Tissue specific expression of starch synthesis-related genes in *A. heterophyllus* and *A. altilis*.

**Table 1 genes-11-00027-t001:** Statistics of the genome assembly of *A. altilis* and *A. heterophyllus*.

	*A. altilis*				*A. heterophyllus*			
Parameters	Contig		Scaffold		Contig		Scaffold	
	Length (bp)	Number	Length (bp)	Number	Length (bp)	Number	Length (bp)	Number
N90	3361	52,085	183,851	637	4902	39,073	77,281	2115
N50	16,898	13,662	1,536,010	151	26,681	9516	547,861	527
N10	47,070	1284	5,076,803	14	82,850	846	1,422,119	54
Total length	803,695,923		833,038,871		930,343,435		982,020,585	
Maximum length	174,221		7,444,155		255,416		3,088,173	
Total number ≥ 100 bp	180,971		98,152		162,440		108,267	
Total number ≥ 2000 bp	61,693		4338		52,444		7263	
N content (%)	3.52				5.26			

**Table 2 genes-11-00027-t002:** BUSCO evaluation of genome assembly of *A. altilis* and *A. heterophyllus*.

BUSCOs	*A. altilis*		*A. heterophyllus*	
	N	P (%)	N	P (%)
Complete BUSCOs	1371	95.20	1369	95.00
Complete single-copy	988	68.60	932	64.70
Complete duplicated	383	26.60	437	30.30
Fragmented	14	1.00	15	1.00
Missing	55	3.80	56	4.00

Abbreviation: BUSCO, Benchmarking universal single-copy orthologs; N, number; P, percentage of complete BUSCOs compared to the total BUSCOs.

**Table 3 genes-11-00027-t003:** The gene coverage based on transcriptome data.

Species	Dataset	Number	Total Length (bp)	Base Coverage by Assembly (%)	Sequence Coverage by Assembly (%)
*A. altilis*	All	141,626	165,794,671	87.7	97.92
	>200 bp	141,626	165,794,671	87.7	97.92
	>500 bp	79,410	146,265,291	86.81	97.62
	>1000 bp	49,485	125,138,638	85.97	96.99
*A. heterophyllus*	All	14,858	6,364,445	90.39	98.89
	>200 bp	14,858	6,364,445	90.39	98.89
	>500 bp	2949	2,853,909	84.41	96.74
	>1000 bp	765	1,386,949	74.83	92.16

**Table 4 genes-11-00027-t004:** Classification of predicted transposable elements in the genome of *A. altilis* and *A. heterophyllus*.

	*A. altilis*		*A. heterophyllus*	
Repeat Type	in Genome (%)	Length (bp)	in Genome (%)	Length (bp)
SINE	0	1187	0.03	384,983
LINE	0.14	1,214,650	0.99	9,775,316
LTR	45.95	382,841,531	36.99	363,293,617
DNA	2.95	24,608,939	3.76	36,982,825
Satellite	0	34,585	0.3	3,001,478
Simple repeat	0.03	253,818	0.04	485,582
Unknown	5.4	45,013,282	12.23	120,128,962
Total	52.04	433,486,547	51.01	500,968,186

**Table 5 genes-11-00027-t005:** Statistics of gene models of *A. altilis*, *A. heterophyllus*, and other species in Rosids.

	*A. altilis*	*A. heterophyllus*	*F. vesca*	*M. domestica*	*M. notabilis*	*P. persica*	*Z. jujuba*
Protein-coding gene number	34,010	35,858	34,301	61,721	27,085	28,701	37,526
Mean gene length (bp)	3545.4	3472.2	2824.6	2692.5	2866.8	2464.8	3313.5
Mean cds length (bp)	1252.6	1241.5	1174.7	1141.4	1086.9	1210.8	1353.0
Mean exons per gene	5.4	5.5	5.1	4.8	4.6	4.9	5.5
Mean exon length (bp)	227.8	226.5	232.5	236.7	236.4	243.6	246.0
Mean intron length (bp)	509.6	497.7	407.1	405.8	494.6	315.8	435.7

**Table 6 genes-11-00027-t006:** Annotation of non-coding RNA genes in the *A. altilis* and *A. heterophyllus* genomes.

Species	Type	Copy (w)	Average Length (bp)	Total Length (bp)	% of Genome
*A. altilis*	miRNA	159	126.7	20,145	0.002418
	tRNA	713	75.3	53,705	0.006447
	rRNA	466	183.2	85,353	0.010246
	18S	76	551.4	41,907	0.005031
	28S	98	125.5	12,296	0.001476
	5.8S	32	135.6	4338	0.000521
	5S	260	103.1	26,812	0.003219
	snRNA	1554	105.4	163,744	0.019656
	CD-box	1410	102.6	144,676	0.017367
	HACA-box	52	130.1	6765	0.000812
	splicing	92	133.7	12,303	0.001477
*A. heterophyllus*	miRNA	168	126.3	21,227	0.002162
	tRNA	689	75.2	51,813	0.005276
	rRNA	2706	268.2	725,709	0.073900
	18S	654	737.5	482,306	0.049114
	28S	920	123.6	113,667	0.011575
	5.8S	242	151.6	36,699	0.003737
	5S	890	104.5	93,037	0.009474
	snRNA	1005	108.2	108,724	0.011071
	CD-box	814	102.5	83,426	0.008495
	HACA-box	68	127.4	8665	0.000882
	splicing	123	135.2	16,633	0.001694

**Table 7 genes-11-00027-t007:** Statistics of functional annotation of protein-coding genes in the *A. altilis* and *A. heterophyllus* genomes.

	*A. altilis*		*A. heterophyllus*	
Values	Number	Percentage	Number	Percentage
Total	34,010	100.0%	35,858	100.0%
Nr	33,353	98.1%	35,076	97.8%
Swissprot	26,689	78.5%	27,741	77.4%
KEGG	24,860	73.1%	25,804	72.0%
COG	12,875	37.9%	13,408	37.4%
TrEMBL	33,240	97.7%	34,968	97.5%
Interpro	26,422	77.7%	27,632	77.1%
GO	17,428	51.2%	18,336	51.1%
Overall	33,394	98.2%	35,109	97.9%
Unannotated	616	1.8%	749	2.1%

## Data Availability

The genome and transcriptome data are deposited in the CNGB Nucleotide Sequence Archive (CNSA: http://db.cngb.org/cnsa; accession number CNP0000715, CNP0000486), and all the annotations are also available via AOCC ORCAE platform (https://bioinformatics.psb.ugent.be/orcae/aocc).

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
