# Peer review of "Draft Genomes of Two Artocarpus Plants, Jackfruit (A. heterophyllus) and Breadfruit (A. altilis)"

_genes, 2019, doi:10.3390/genes11010027_

Round 1
Reviewer 1 Report
The manuscript by Sahu et al. described genome assemblies and -analyses of jackfruit (A. heterophyllus) and breadfruit (A. altilis). The assembly is highly fragmented most probably due to their heterozygous genomes and the strategy adopted for sequencing. They constructed a number of paired-end and mate-pair libraries for each genome and used only Platanus tool for assembly. I would recommend to use two other assembly tools (eg. MaSuRCA and ALLPATH-LG) to check whether Platanus provided the best assembly or not. 'Genome evaluation' should be 'Genome assembly evaluation'. 'Genome assembly' must be mentioned while describing analyses of genomes. The manuscript did not mention the accession numbers of the varieties of jackfruit and breadfruit they used for sequencing. I have a concern about the method adopted for genome size estimation. The two peaks appeared in the K-mer distribution curve (Fig. S1) appeared to be due to heterozygous genomes. In that case, K-mers under both the peaks need to be considered for genome size estimation (Ramos et al., 2018 Scientific Data). Supplementary figure legends should be elaborated for understanding of non-experts. Distribution of scaffolds/contigs sizes should be shown as a bar diagram as shown in figure S2A. A syntenic analysis between two largest orthologous scaffolds of jackfruit and breadfruit may be presented with diagram, so that gene orientation and intergenic distances between two species can be compared. A comparison of starch contents of the sequenced accessions of the two species may be determined can correlated with the data presented in figure 4. The figure S3A is presented in a wrong manner. Some grammatical mistakes and typo errors should be corrected.
Author Response
The manuscript by Sahu et al. described genome assemblies and -analyses of jackfruit (A. heterophyllus) and breadfruit (A. altilis). The assembly is highly fragmented most probably due to their heterozygous genomes and the strategy adopted for sequencing. They constructed a number of paired-end and mate-pair libraries for each genome and used only Platanus tool for assembly. I would recommend to use two other assembly tools (eg. MaSuRCA and ALLPATH-LG) to check whether Platanus provided the best assembly or not.
Response: Thank you for the valuable suggestion. We now used the GCE software to evaluate the heterozygosity, and the results showed that the heterozygous ratio is 1.13% and 0.911% for A. altilis and A.heterophyllus, respectively. As Platanus is highly efficient for the assembly of highly heterozygous genomes [1], and based on our prior experience with several heterozygous genomes, we selected Platanus for the genome assembly.
Fu, Y.; Li, L.; Hao, S.; Guan, R.; Fan, G.; Shi, C.; Wan, H.; Chen, W.; Zhang, H.; Liu, G., et al. Draft genome sequence of the Tibetan medicinal herb Rhodiola crenulata. GigaScience 2017, 6, doi:10.1093/gigascience/gix033.
'Genome evaluation' should be 'Genome assembly evaluation'. 'Genome assembly' must be mentioned while describing analyses of genomes.
Response: 'Genome evaluation' is changed to 'Genome assembly evaluation'. The description about genome assembly is added in the result section.
The manuscript did not mention the accession numbers of the varieties of jackfruit and breadfruit they used for sequencing.
Response: The genome and transcriptome data are deposited in the CNGB Nucleotide Sequence Archive (CNSA: http://db.cngb.org/cnsa; accession number CNP0000715, CNP0000486), and all the annotations are also available via AOCC ORCAE platform (https://bioinformatics.psb.ugent.be/orcae/aocc). The genome data will be made available after acceptance of the manuscript. However now we have provided genome portals for these species via ORCAE database of AOCC:
- Breadfruit: https://bioinformatics.psb.ugent.be/orcae/aocc/overview/Artal
- Jackfruit: https://bioinformatics.psb.ugent.be/orcae/aocc/overview/Arthe
They’re currrently being restricted genomes but the reviewer can login with credential: User: RV and Password: 9BPQ Sequences and annotation files can be directly downloaded in the “Download” section.
I have a concern about the method adopted for genome size estimation. The two peaks appeared in the K-mer distribution curve (Fig. S1) appeared to be due to heterozygous genomes. In that case, K-mers under both the peaks need to be considered for genome size estimation (Ramos et al., 2018 Scientific Data).
Response: We have used two methods for estimating the genome size, 17mer [2] and GCE, the second peaks was the homozygous peak, and the first peaks was the heterozygous peak [2], and we also found that our estimated genome size of A. altilis and A. heterophyllus was relatively close to the genome size of species in the genus Artocarpus based on existing data in the C-values database (https://cvalues.science.kew.org/search/angiosperm), where 1C-value is 1.2 pg.
|
A. altilis |
the genome size of heterozygous peak(bp) |
the genome size of homozygous peak(bp) |
|
17mer-analysis |
1,578,874,146 |
811,259,670 |
|
GCE |
864,925,000 |
|
|
A. heterophyllus |
the genome size of heterozygous peak(bp) |
the genome size of homozygous peak(bp) |
|
17mer-analysis |
1,928,998,464 |
1,005,609,005 |
|
GCE |
979,768,000 |
|
Teh, B.T.; Lim, K.; Yong, C.H.; Ng, C.C.Y.; Rao, S.R.; Rajasegaran, V.; Lim, W.K.; Ong, C.K.; Chan, K.; Cheng, V.K.Y., et al. The draft genome of tropical fruit durian (Durio zibethinus). Nature Genetics 2017, 49, 1633-1641, doi:10.1038/ng.3972.
Supplementary figure legends should be elaborated for understanding of non-experts.
Response: We have now added additional information in Figure S1 and Figure S4 for understanding of non-experts.
Distribution of scaffolds/contigs sizes should be shown as a bar diagram as shown in figure S2A.
Response: We have now added the distribution of scaffolds sizes in Figure S3.
A syntenic analysis between two largest orthologous scaffolds of jackfruit and breadfruit may be presented with diagram, so that gene orientation and intergenic distances between two species can be compared.
Response: We have now used mcscan to find the collinearity between two genomes, then selected the largest 50 orthologous scaffolds to show the collinearity (Figure S7).
A comparison of starch contents of the sequenced accessions of the two species may be determined can correlated with the data presented in figure 4.
Response: Unfortunately, we do not have starch content data at the moment. We will consider this suggestion in our subsequent manuscript that will provide a more detailed investigation of the biology of the plant, along with some additional experiments.
The figure S3A is presented in a wrong manner.
Response: Thank you for pointing out the mistake, we have modified it accordingly.
Some grammatical mistakes and typo errors should be corrected.
Response: We have now thoroughly revised the manuscript to avoid the grammatical and typo errors. The MS has been also proof read by native English-speaking authors of this manuscript.
Reviewer 2 Report
Sahu et al., describe two draft genomes for jackfruit and breadfruit. The authors present a pretty standard genome style paper with a description of the assembly, gene prediction, gene family analysis, whole genome duplication analysis, and then finish off with an expression analysis. From the data presented most methods seem to be technically sound. The genomes are based on Illumina PE and MP, which provides high quality draft assemblies for gene space. This is evidenced by their good BUSCO scores. However, the contiguity is low compared to many genomes being published now based on long read datasets. Also, based on the kmer plots these are heterozygous genomes. What is the level of heterozygosity? This is also seen in their BUSCO scores with 26 and 30% duplicated genes so some of the downstream gene analysis (WGD and overrepresented families) might be mis-leading based on collapsed and non-collapsed (haplotype) fragments in the genomes. The authors also don’t try any scaffolding technologies, such as HiC, Bionano or tried and true genetic maps, which are also more standard for genome drafts currently. So overall the genomes will be a great resource for gene discovery for the community and are important for some aspects of molecular breeding of these orphan crops. However, the genomes were not available at either of the places mentioned in the manuscript, so they could not be evaluated by this reviewer and are not available for the community as a resource. Many genomes are being published that are not really available to the community, so if this is to be a true resource then they would need to be available to evaluate during the review process, as well as validate that the data that is claimed to be archived is there.
Author Response
Reviewer 2
Comments and Suggestions for Authors
Sahu et al., describe two draft genomes for jackfruit and breadfruit. The authors present a pretty standard genome style paper with a description of the assembly, gene prediction, gene family analysis, whole genome duplication analysis, and then finish off with an expression analysis. From the data presented most methods seem to be technically sound. The genomes are based on Illumina PE and MP, which provides high quality draft assemblies for gene space. This is evidenced by their good BUSCO scores. However, the contiguity is low compared to many genomes being published now based on long read datasets.
Response: Thank you for the constructive suggestions. Please find below our specific responses.
Also, based on the kmer plots these are heterozygous genomes. What is the level of heterozygosity? This is also seen in their BUSCO scores with 26 and 30% duplicated genes so some of the downstream gene analysis (WGD and overrepresented families) might be mis-leading based on collapsed and non-collapsed (haplotype) fragments in the genomes.
Response: Thank you for the nice suggestion. We now used the GCE software to evaluate the heterozygosity, and the results showed that the heterozygous ratio is 1.13% and 0.911% for A. altilis and A.heterophyllus, respectively. The high heterozygous ratio possibly resulted in higher number of duplicated genes.
The authors also don’t try any scaffolding technologies, such as HiC, Bionano or tried and true genetic maps, which are also more standard for genome drafts currently. So overall the genomes will be a great resource for gene discovery for the community and are important for some aspects of molecular breeding of these orphan crops.
Response: We totally understand your concerns. As this is our preliminary yet first stage of genomic analyses of these Atrocarpus genomes, we have not performed chromosome level assemblies. However, this will be incorporated in our follow up studies, along with several experimental analyses.
However, the genomes were not available at either of the places mentioned in the manuscript, so they could not be evaluated by this reviewer and are not available for the community as a resource. Many genomes are being published that are not really available to the community, so if this is to be a true resource then they would need to be available to evaluate during the review process, as well as validate that the data that is claimed to be archived is there.
Response: The genome data will be made available after acceptance of the manuscript. However now we have provided genome portals for these species via ORCAE database of AOCC:
- Breadfruit: https://bioinformatics.psb.ugent.be/orcae/aocc/overview/Artal
- Jackfruit: https://bioinformatics.psb.ugent.be/orcae/aocc/overview/Arthe
They’re currrently being restricted genomes but the reviewer can login with credential: User: RV and Password: 9BPQ
Sequences and annotation files can be directly downloaded in the “Download” section.
Reviewer 3 Report
Genes Review Sahu et al. describe their assembly and annotation of jackfruit and breadfruit genomes in their manuscript “Draft genomes of two Artocarpus plants, Jackfruit and Breadfruit”. This research is important and will support tropical fruit research that is lagging behind many temperate species. The analyses are appropriate for their data. While the genomes are highly fragmented, the authors do not overstate their findings and their results significantly advance our understanding. Overall, this is an important contribution to the scientific community. Specific requested revisions follow. L88 Provide more information on the cultivars/varieties/accessions sequenced. These must have some accession identifier that can be used to distinguish them. This is important for any follow up work. L88 Please state the availability of the sequenced accessions for distribution for others to build upon your work. L95 State the reason why there is a difference between Q score cutoffs between the two species. L99 “their estimated” L99 State the genome sizes briefly in the text with a reference (flow cytometry estimates?). State if this is not currently known. L128 This section is awkward as written. Please correct by including information in paragraphs with multiple, connected sentences per paragraph. L131 State what “only significant difference” refers to. This sentence is better placed in the results/discussion. L 169 This section is awkward as written. Please correct by including information in paragraphs with multiple, connected sentences per paragraph. L242 “amino acid protein sequences”? State the database source and IDs of the bait sequences (sequence IDs can be placed in supplemental data). L261 It is not usually necessary to include all N10-N90 data. N50 and N90 are most common. You can save space by eliminating additional lines in this table. L276 Define use of P in “P (%)”. Also, clarify if “N, number” is for “No.” column header in table. If not, clarify. L292 Add space to “% in genome”. L386 Figure 4 should have a color key.
Author Response
Comments and Suggestions for Authors
Genes Review Sahu et al. describe their assembly and annotation of jackfruit and breadfruit genomes in their manuscript “Draft genomes of two Artocarpus plants, Jackfruit and Breadfruit”. This research is important and will support tropical fruit research that is lagging behind many temperate species. The analyses are appropriate for their data. While the genomes are highly fragmented, the authors do not overstate their findings and their results significantly advance our understanding. Overall, this is an important contribution to the scientific community. Specific requested revisions follow.
Response: Thank you. The comments were highly insightful and enabled us to greatly improve the quality of our manuscript. We thoroughly revised the manuscript, especially the introduction part, and removed the wrong references and replaced them with the correct and updated ones.
L88 Provide more information on the cultivars/varieties/accessions sequenced. These must have some accession identifier that can be used to distinguish them. This is important for any follow up work.
Response: We have now added the accession details in the main text as follows “Genomic DNA was extracted from fresh leaves of A. heterophyllus (ICRAFF 11314) and A. altilis (ICRAFF 11315), grown at the World AgroForestry (ICRAF) campus in Kenya, using a modified CTAB method [33].
L88 Please state the availability of the sequenced accessions for distribution for others to build upon your work.
Response: The genome and transcriptome data are deposited in the CNGB Nucleotide Sequence Archive (CNSA: http://db.cngb.org/cnsa; accession number CNP0000715, CNP0000486), and all the annotations are also available via AOCC ORCAE platform (https://bioinformatics.psb.ugent.be/orcae/aocc). The genome data will be made available after acceptance of the manuscript. However now we have provided genome portals for these species via ORCAE database of AOCC:
- Breadfruit: https://bioinformatics.psb.ugent.be/orcae/aocc/overview/Artal
- Jackfruit: https://bioinformatics.psb.ugent.be/orcae/aocc/overview/Arthe
They’re currrently being restricted genomes but the reviewer can login with credential: User: RV and Password: 9BPQ Sequences and annotation files can be directly downloaded in the “Download” section.
L95 State the reason why there is a difference between Q score cutoffs between the two species.
Response: As the quality of the mate-paired library of A. heterophyllus was not good compared to A. altilis, a different Q score cutoffs were used for each species.
L99 “their estimated” L99 State the genome sizes briefly in the text with a reference (flow cytometry estimates?). State if this is not currently known.
Response: The genome size estimates has been added in the results section. We have used two methods for estimating the genome size, 17mer [2] and GCE, the second peaks was the homozygous peak, and the first peaks was the heterozygous peak [2], and we also found that our estimated genome size of A. altilis and A. heterophyllus was relatively close to the genome size of species in the genus Artocarpus based on existing data in the C-values database (https://cvalues.science.kew.org/search/angiosperm), where 1C-value is 1.2 pg.
Teh, B.T.; Lim, K.; Yong, C.H.; Ng, C.C.Y.; Rao, S.R.; Rajasegaran, V.; Lim, W.K.; Ong, C.K.; Chan, K.; Cheng, V.K.Y., et al. The draft genome of tropical fruit durian (Durio zibethinus). Nature Genetics 2017, 49, 1633-1641, doi:10.1038/ng.3972.
L128 This section is awkward as written. Please correct by including information in paragraphs with multiple, connected sentences per paragraph.
Response: The sentence has been revised as “The genome assembly completeness was assessed using BUSCO (Benchmarking Universal Single-Copy Orthologues), version 3.0.1 (BUSCO, RRID:SCR_015008) [18]. Next, the unigenes generated by Bridger software [19] from the transcriptome data of each species were aligned to the assembled genomes using BLAT (BLAT, RRID:SCR_011919) [20] with default parameters. Then, in order to confirm the accuracy of the assembly, some of the paired-end libraries (170, 250 and 350 bp) were aligned to the assembled genomes, and the sequencing coverage was calculated using SOAPaligner, version 2.21 (SOAPaligner/soap2, RRID:SCR_005503) [21].”
L131 State what “only significant difference” refers to. This sentence is better placed in the results/discussion.
Response: The sentence has been revised and moved to results as “We observed a significant difference in the number of duplicated core genes in A. altilis and A. heterophyllus [Table 2], which might be probably due to the genome duplication in these species.”
L 169 This section is awkward as written. Please correct by including information in paragraphs with multiple, connected sentences per paragraph.
Response: The sentence has been revised as “Repetitive regions of the genome were masked before gene prediction. Based on the RNA, homologous and de novo prediction evidences the protein-coding genes were identified using the MAKER-P pipeline (version 2.31) [29]. For RNA evidence, the clean transcriptome reads were assembled into inchworms using Trinity version 2.0.6 [30], and then fed to MAKER-P as EST evidence. For homologous evidence, the protein sequences from four relative species in rosids (F. vesca, M. domestica, M. notabilis, Prunus persica, Ziziphus jujuba) were downloaded and provided as protein evidence.”
L242 “amino acid protein sequences”? State the database source and IDs of the bait sequences (sequence IDs can be placed in supplemental data).
Response: We have now included the suggested information and added a table S8 in the additional files.
L261 It is not usually necessary to include all N10-N90 data. N50 and N90 are most common. You can save space by eliminating additional lines in this table.
Response: We modified the statistics of Table 1 as per the suggestions.
L276 Define use of P in “P (%)”. Also, clarify if “N, number” is for “No.” column header in table. If not, clarify.
Response: We have modified the table and foot note as per the suggestions. “N, number; P, percentage of complete BUSCOs compared to the total BUSCOs”
L292 Add space to “% in genome”.
Response: added
L386 Figure 4 should have a color key.
Response: Color key is added now.
Round 2
Reviewer 1 Report
The revised manuscript is improved and nothing much can be done with this state of assembly. By the 'accession number' I wanted to mean the germplasm (variety) accession numbers of the two species. This would define which specific accessions of the species they have sequenced. I am not satisfied with the syntenic analysis (Fig.S7) they have performed. This is just a nucleotide level comparison. I wrote 'A syntenic analysis between two largest orthologous scaffolds of jackfruit and breadfruit may be presented with diagram, so that gene orientation and intergenic distances between two species can be compared'. So, a protein and gene level analysis using MCScanX is required to address the issue. They should do whatever can be done with this assembly.
Author Response
The revised manuscript is improved and nothing much can be done with this state of assembly. By the 'accession number' I wanted to mean the germplasm (variety) accession numbers of the two species. This would define which specific accessions of the species they have sequenced.
Response: Thank you for the positive feedback. We do understand your concern regarding the accession number. The two names used are the official accession numbers (A. heterophyllus (ICRAFF 11314) and A. altilis (ICRAFF 11315), thus the names from the World Agroforestry Centre (ICRAF) GERMPLASM BANK in Nairobi where they were sampled. These species details can easily be obtained by other researchers based on the given accession number.
I am not satisfied with the syntenic analysis (Fig.S7) they have performed. This is just a nucleotide level comparison. I wrote 'A syntenic analysis between two largest orthologous scaffolds of jackfruit and breadfruit may be presented with diagram, so that gene orientation and intergenic distances between two species can be compared'. So, a protein and gene level analysis using MCScanX is required to address the issue. They should do whatever can be done with this assembly.
Response: Thank you for the suggestion. We have now revised the Fig. S7. The collinearity of the largest orthologous scaffolds were determined and plotted using MCscan and the JCVI utility libraries v0.9.14. The result indicated the conserved shared synteny. In addition, we also revised Fig. 1d for more clarity.
